# Zooming in and out: Exploring RNA Viral Infections with Multiscale Microscopic Methods

**DOI:** 10.3390/v16091504

**Published:** 2024-09-23

**Authors:** Cheng-An Lyu, Yao Shen, Peijun Zhang

**Affiliations:** 1Division of Structural Biology, Wellcome Centre for Human Genetics, University of Oxford, Oxford OX3 7BN, UK; cheng-an.lyu@ndm.ox.ac.uk; 2Chinese Academy of Medical Sciences Oxford Institute, University of Oxford, Oxford OX3 7BN, UK; 3Diamond Light Source, Harwell Science and Innovation Campus, Didcot OX11 0DE, UK

**Keywords:** RNA virus, viral infection, virus–host interactions, multiscale microscopic methods, cryo-soft X-ray tomography (cryo-SXT), serial cryo-focused ion beam/scanning electron microscopy cryo-FIB/SEM volume imaging, cellular cryo-tomography (cryo-ET), cryo-EM single-particle analysis (SPA)

## Abstract

RNA viruses, being submicroscopic organisms, have intriguing biological makeups and substantially impact human health. Microscopic methods have been utilized for studying RNA viruses at a variety of scales. In order of observation scale from large to small, fluorescence microscopy, cryo-soft X-ray tomography (cryo-SXT), serial cryo-focused ion beam/scanning electron microscopy (cryo-FIB/SEM) volume imaging, cryo-electron tomography (cryo-ET), and cryo-electron microscopy (cryo-EM) single-particle analysis (SPA) have been employed, enabling researchers to explore the intricate world of RNA viruses, their ultrastructure, dynamics, and interactions with host cells. These methods evolve to be combined to achieve a wide resolution range from atomic to sub-nano resolutions, making correlative microscopy an emerging trend. The developments in microscopic methods provide multi-fold and spatial information, advancing our understanding of viral infections and providing critical tools for developing novel antiviral strategies and rapid responses to emerging viral threats.

## 1. Introduction

RNA viruses represent a diverse and highly adaptive group of pathogens that play a central role in shaping the landscape of biology and human health [1]. RNA viruses use ribonucleic acid (RNA) as their genetic material and can be classified into single-stranded RNA (ssRNA) viruses, double-stranded RNA (dsRNA) viruses, and retroviruses, depending on their genome type and replication strategy. Single/double-stranded RNA viruses, such as influenza, SARS-CoV-2, and Ebola, rely on RNA-dependent RNA polymerases for genome replication. In contrast, retroviruses such as the human immunodeficiency virus (HIV) employ reverse transcriptase to produce viral DNA, which is then integrated into the host genome by integrase. RNA viruses are characterized by high mutation rates and rapid evolution, allowing them to quickly escape host immune responses and develop resistance to antiviral treatments. This adaptability underscores the significant public health threats posed by RNA viruses, as evidenced by recent outbreaks and pandemics caused by the Ebola virus and SARS-CoV-2 [2,3].

Continued research into RNA viruses remains crucial for understanding their pathology and has facilitated breakthroughs in vaccine development, antiviral treatments, and our comprehension of the immune system [4,5,6]. The advancement of various imaging techniques, notably electron microscopy (EM), has proven robust in delineating complex structures of RNA viruses, viral life cycles, exploring virus–host interactions, and unravelling host immune responses [7,8]. This review aims to provide a comprehensive overview of cryo-imaging techniques that offer multiscale insights into RNA virus biology that allow for a holistic view of virus-infected cells in a native, hydrated state. These cryo-techniques, organized by observation scale from large to small, include cryo-soft X-ray tomography (cryo-SXT), serial cryo-focused ion beam/scanning electron microscopy cryo-FIB/SEM volume imaging, cellular cryo-tomography (cryoET), and cryo-electron microscopy (cryo-EM) single-particle analysis (SPA) (Figure 1). When used together, they can provide unprecedented details of ultrastructural features of virus-infected cells, and contextual information that bridges the gap between molecular mechanisms and cellular responses. In this paper, we emphasize the importance of microscopy techniques in RNA virus research and showcase how these diverse imaging modalities can be used synergistically for comprehensive studies of RNA viral infection. We summarize the achievable resolutions of these techniques in Figure 1, followed by their respective workflows in Figure 2, and applications in Figure 3 and Figure 4.

## 2. Seeing through the Whole Cell: Cryo-Soft X-ray Tomography

Cryo-SXT has emerged as a robust tool to image whole intact cells in a high throughput manner, ideally for characterizing cells upon viral infection that can provide unique insights into the ultrastructural details of virus-infected cells. Compared with conventional X-ray tomography, it uses low-energy X-rays, typically in the “water window” range of 284 to 543 eV, to image hydrated, intact cells, without the need for chemical fixing, staining or sectioning [13,14,15]. Under this condition, carbon-rich biological samples produce high contrast while oxygen-rich structures such as the cytosol remain virtually transparent, enabling the visualization of cellular compartments. Despite its slightly lower achievable resolution compared to electron microscopy, about 25 to 50 nanometers depending on the zone plate, cryo-SXT excels at imaging relatively thick samples (up to 15 μm) rapidly. It can capture an entire three-dimensional 3D volume of 15 × 15 × 15 µm^3^ in about 5 min and allow the examination of large, complex cellular structures and significant ultrastructural alterations caused by infection [14,16]. Moreover, when integrated with fluorescence microscopy, viral particles can be localized for targeted imaging, enabling the characterization of viral trafficking and viral life cycle in cells [17]. These distinctive features of cryo-SXT make it a robust approach for investigating RNA virus infections, offering details for cellular response to viral infection, viral life cycle, and providing details that are not accessible with other imaging modalities.

The cryo-SXT workflow is a systematic process, in which cells are seeded or deposited on sterilized microscopy grids or housed in thin-walled glass capillary tubes. For virus research, cells are inoculated with the virus and incubated under appropriate biosafety conditions. Vitrification is performed by high-pressure freezing (HPF) or plunge freezing, with gold particles or fluorescent stains added for tomogram alignment and data analysis. In the transmission X-ray microscope (TXM), these grids are first imaged in visible light to select regions of interest (ROI) for X-ray imaging. Tomograms of ROIs are acquired by capturing a series of transmission images at various tilt angles of the sample to capture structural information at different orientations (Figure 2). Depending on the samples, the angular range can be up to 140 degrees for EM grids and 360 degrees for capillary tubes through the entire cell with a thickness up to 16 μm [18]. Image stacks are normalized and aligned using the established processing workflow such as AREC3D [19,20,21,22], and the reconstructed volumes are produced for detailed analysis.

Leveraging cryo-SXT, there has been an increasing amount of research in various RNA virus infections [10,18,23,24,25,26,27,28]. We herein take the research on hepatitis C virus (HCV) and reovirus as examples to illustrate the contribution of cryo-SXT to the research of RNA viruses. HCV is an enveloped positive-strand RNA (+ssRNA) virus notorious for its ability to induce serious liver damage [29]. To characterize the cell response to HCV infection and antiviral treatment, Pérez-Berná et al. [9,27] obtained 3D maps of the whole cells harboring HCV replicons at different stages of infection, as well as after the treatment of antiviral drugs using cryo-SXT. In Figure 3a, the HCV replicon-bearing cell slices are shown, with projections (Figure 3a(i,ii)) and volume slices (Figure 3a(iii,iv)) providing detailed views of the selected ROI and identified organelles. In their research, significant changes in the structure of the endoplasmic reticulum (ER) and mitochondria were seen in nearly native, whole, hydrated cells during HCV infection and antiviral treatments. HCV replication relies on special networks in HCV-infected cells formed by vesicular and tubular structures derived from the ER. Meanwhile, abnormal mitochondria were observed, which indicates the mitochondrial dysfunction induced by the virus. Notably, these morphological changes were reverted by clinically approved antivirals sofosbuvir and daclatasvir. These results suggest that the cellular morphology observed by cryo-SXT can be used as a criterion for evaluating the effectiveness of other potential antiviral drugs. Their study highlights the promise of this method to aid in drug development during the preclinical stage, which allows for the assessment of the possible influence of candidate compounds on the cellular ultrastructure.

In parallel, reovirus is an icosahedral, non-enveloped dsRNA virus that can be potentially applied to oncolytic therapy [30]. Non-enveloped viruses do not have the membrane-fusion ability, making their cell-entry process largely unknown. To track the entry process of reovirus and modification of endosomes from early to late stages post-infection, Kounatidis et al. developed a correlative microscopy system combining cryo-SXT and other techniques and achieved direct data correlation in 3D [24]. By combining cryo-SXT with 3D-structure illumination microscopy (3D-SIM), the study provided high-resolution 3D views of the cellular compartments involved in viral trafficking, allowing for the visualization of vesicles carrying viral particles and their transformations over time. This approach enabled the identification of multi-vesicular bodies and other vesicle structures that form during infection, providing insights into how reoviruses escape from endosomes into the cytoplasm without disrupting the overall vesicle integrity. These findings underscore the utility of cryo-SXT in capturing the dynamic interactions between reoviruses and host cell organelles, contributing to a more comprehensive understanding of viral pathogenesis.

These studies demonstrate that cryo-SXT is effective in the characterization of virus–host interactions, from imaging viral particles within infected cells to investigating changes to cell ultrastructure and organelles post-infection to facilitate the development of future treatments.

## 3. Peeling Back the Layers: Serial Cryo-FIB/SEM Volume Imaging

Another rapidly evolving imaging technique allowing whole-cell imaging is serial cryo-focused ion beam/scanning electron microscopy (cryo-FIB/SEM) volume imaging. This technique allows the visualization of relatively thick frozen-hydrated biological specimens (e.g., virus-infected cells) at resolutions ~10 nm. It employs a dual-beam, enabling the serial sectioning of thick samples (above 300 nm up to hundreds of microns) using a focused ion beam coupled with subsequent imaging with SEM. Briefly, biological samples (e.g., cells and tissues) that have been either plunge-frozen or high-pressure-frozen undergo serial sectioning using a focused ion beam, typically composed of gallium ions or plasma ions to mill away a thin slice of the sample. The milling process is regulated in a way to allow for precise removal of thin slices of material ranging from 3 to 15 nm. Immediately following the milling, the sample surface is imaged by collecting backscattered and secondary electrons, using an integrated scanning electron beam to capture high-resolution details about the sample. The sequential cryo-focused ion beam (cryo-FIB) milling, and SEM imaging are repeated iteratively (up to hundreds to thousands of times) until a series of images covering the entire volume of interest is obtained. Due to these repeated steps, imaging of an entire mammalian cell can take up to days. The collected 2D images are then computationally reconstructed into a 3D volume for the whole cell, providing details into the spatial organization of cellular compartments, the 3D architecture of cellular structures, such as organelles, membranes, and other subcellular components in their native, hydrated state. Moreover, this technique can also be combined with florescent microscopy [31] for cryo-FIB lamella preparation to target a region of interest, allowing for penetration of electron beam imaging using cryo-ET to obtain high-resolution information of the targeted region, which will be discussed in the following section.

Given its straightforward sample preparation process, serial cryo-FIB/SEM volume imaging has been successfully applied to various biological samples ranging from cells to multicellular organisms and tissues [32,33,34,35,36,37,38]. Its capability to reveal the spatial organization of cellular compartments has made it increasingly popular for studying viral infections, with a notable surge in research on SARS-CoV-2 in recent years [10,39].

SARS-CoV-2 coronaviruses are small, enveloped viruses containing a single-stranded, positive RNA genome. While the viral components have been extensively characterized to an atomic resolution [40,41], understanding of the SARS-CoV-2 coronavirus infection at the cellular level is limited. Using a multi-scale imaging technique including volume imaging using cryo-SXT, serial cryo-FIB/SEM volume imaging, and cellular tomography imaging (explained in the next section), Mendonca et al. [10] provided a comprehensive characterization of SARS-CoV-2 infection within frozen hydrated Vero cells (Figure 3b). Volume imaging reveals drastic cytopathic alterations of SARS-CoV-2-infected cells, including both mitochondrial defects in morphology and number, as well as nucleus defects resulting from cytoplasmic invagination (Figure 3b(i)). In addition to the visualization of the cellular compartment, virus particles represented by electron-dense particles can be unambiguously identified at the cell periphery, inside the cytoplasm and within vesicles (Figure 3b(i–iii)), highlighting the robustness of volume imaging in virus research. Combined with cryo-ET imaging on cell lamella (explained in the next section), this work shed light on the SARS-CoV-2 life cycle from viral RNA replication and transport to virus assembly and egress within a native cellular context, providing a holistic view of the SARS-CoV-2 infection within cells that could not have been possible without the combination of different imaging techniques.

## 4. Zooming in on a Single Layer: Cellular Cryo-Electron Tomography

Cryo-ET represents one of the most important imaging techniques in structural biology that bridges the resolution gap between volumetric imaging and high-resolution structural determination methods such as SPA and X-ray crystallography. Unlike the later methods, which excel in revealing atomic or near-atomic details of isolated macromolecules, cryo-ET provides a unique window to characterize macromolecular complexes within their natural cellular environments. Cryo-ET is most effective when imaging samples thinner than 300 nm, thus can be directly used for biological samples, including purified virions and viruses located within thin regions such as the cell periphery [10,42]. For thicker samples, such as whole cells, where electron penetration is insufficient, cryo-FIB can be used to prepare thin lamellae of 80–250 nm thickness and subsequently imaged using cryo-ET. This combination allows imaging capabilities to extend deep into the cell, enabling the characterization of complete viral life cycles, their interactions with host cells, and the cellular immune response during viral infection.

Cryo-ET requires capturing a series of two-dimensional projections of a specimen using a microscope stage that is tilted at different angles in relation to the incident beam [43,44,45]. The concatenate projections are widely referred to as the ‘tilt-series’, spanning an angular range of ±60° with increments of a few degrees per image of the target region. Due to the mechanical properties of the stage and the limited tilt angle used, cryo-ET imaging often leads to incomplete sampling of Fourier space, leading to the so called “missing wedge” problem [46]. To minimize radiation damage while retaining high-resolution information, several data collection strategies can be applied, including continuous, bidirectional, and dose-symmetric tilt schemes. The dose-symmetric tilt scheme, which starts at a low angle and alternates between progressively positive and negative angles, is the most used cryo-ET data collection strategy due to its optimized, near-symmetric information transfer [47]. Post-collection, the tilt series undergo motion correction and alignment using various software packages (e.g., emClarity [48,49], RELION [50], IMOD [51]) to generate tomograms and offer a three-dimensional depiction of the target region on the biological specimens [52,53,54,55].

Collecting a group of tomograms, or sometimes even single tomograms, can offer a thorough examination of biological processes. Moreover, when tomograms contain multiple copies of a macromolecule of interest, subtomogram averaging (STA) [43,48,49,50] can achieve near-atomic resolution structures by aligning, extracting, and merging multiple segments from the tomograms, enhancing the signal-to-noise ratio for specific features such as viral proteins or ribonucleoprotein complexes (RNP) [56,57]. Together, cryo-ET combined with STA is powerful for studying diverse viral structures with a high level of detail [48,58].

Advancements in cryo-ET methods over the past two decades have provided valuable insights into viral assemblies and viral protein structures from in vitro isolated virions, including those of the Ebola virus [59,60], Marburg virus [60], HIV-1 [61], and SARS-CoV-2 [57,62,63], facilitating the rapid response to viral outbreaks and vaccine development. Recently, developments in cellular cryo-ET have enabled the characterization of virus particles and vaccine candidates within cells to a comparable resolution, as well as their interactions with host cells [10,11,41,64,65,66]. This is exemplified by studies on bluetongue virus (BTV) and mumps virus (MuV) infection, respectively.

Bluetongue virus (BTV) is a dsRNA virus in the family of Reoviridae that can cause diarrhea in humans and death to livestock [67]. The BTV virion consists of three layers of capsid protein encapsulating ten segments of dsRNA genome. Despite the well-known structures of the BTV particles, the genome packaging and viral assembly process remain unclear. Leveraging in-cell cryo-ET advancement in studies of reovirus and rotavirus infection processes [68,69], Xia et al. [64] captured six assembly intermediates of BTV to intermediate resolutions (up to 7.2 Å) in infected baby hamster kidney (BHK21) cells. By combining these structures with high-resolution SPA of isolated viral particles, they demonstrated a comprehensive process of virus assembly. This process begins with the co-assembly of newly synthesized ssRNA segments with the first layer of capsid (viral protein VP6 and VP3) in the virus inclusion body, followed by the sequential assembly of the outer two capsid layers (second layer VP7 and third layer VP2), and continues with the egress of immature virus particles mediated by a virus-encoded transmembrane protein and the outermost capsid (VP2).

The integration of cryo-correlative light and electron microscopy (cryo-CLEM) with cryo-ET has proven to be highly effective for characterizing viral life cycles in cells. Using a cryo-correlative imaging workflow, Zhang et al. [11] investigated the reactivation mechanism of MuV replication under oxidative stress conditions in long-term infected HeLa cells. They prepared thin cell lamellae containing MuV factories by targeting the closely located and fluorescently labeled G3BP stress granule assembly factor 1 (G3BP1) (Figure 4a). Cryo-ET imaging revealed altered MuV nucleocapsid morphologies under oxidative stress conditions, with an increased abundance of straight filaments (Figure 4b). Using in-cell subtomogram averaging, they achieved a reconstruction of the MuV capsid at 6.5 Å resolution, which adopted a genome-accessible conformation (Figure 4c). This conformation, along with additional densities in the viral capsid, elucidates how partner proteins phosphoprotein P and polymerase L access the viral RNA to activate replication. This work sheds light on how MuV–host interplay disrupts the metastable balance during long-term infection under host stress that allows the activation of viral replication and exemplifies a robust cryo-correlative workflow to study other RNA infections.

## 5. Spotlight on Key Components: Single-Particle Analysis

SPA is the most established cryo-EM technique for determining the structures of macromolecules in their near-native state at near-atomic resolution [70]. SPA conventionally does not provide information on viral particles within cells, but it does provide high resolutions that are not attainable with other microscopic imaging techniques. Nevertheless, a recent trend has emerged where SPA technique is being utilized to analyze protein particles in thin lamellae [71].

SPA workflow begins with preparing and rapidly freezing in vitro purified or assembled macromolecules to preserve particles in a thin aqueous layer (<150 nm). Subsequently, images containing projections of randomly oriented particles are recorded using an electron microscope and direct detection cameras are then processed computationally to generate 3D reconstructions of complexes. Specifically, individual particles are detected, aligned, and classified into groups based on their projections, enhancing signal and reducing noise. A three-dimensional model is reconstructed from these 2D images consisting of varied angles of particle orientation. This initial model undergoes iterative 3D classification, refinement, and validation to ensure accuracy in the angular assignment and resolution. SPA is particularly valuable for studying flexible or dynamic large protein assemblies with a high degree of heterogeneity (compositional and conformational). Routine and automatic data collection enables efficient data collection (5000 micrographs per day). These large amounts of data available can provide near-atomic resolution of RNA virus components such as the viral capsid [72,73], viral envelope, spikes, and viral replication machinery [74] and facilitate vaccine and small molecule therapeutics development [75]. Notably, viral components like viral capsid which were traditionally imaged using cryo-ET have recently been characterized using SPA, resulting in higher resolution and novel mechanistic insights [12,76,77].

The work of Schirra et al. on the HIV-1 capsid [12] illustrates the power of SPA in revealing novel molecular mechanisms. In Figure 4d, in vitro-assembled VLPs with declinations encircled are shown, while Figure 4e depicts the map after further local refinement with imposed five-fold symmetry, colored according to local resolution. Their study identified a Thr-Val-Gly-Gly motif in the viral capsid protein (CA) that acts as a molecular switch, controlling the formation of hexamers and pentamers in the fullerene capsid structure. This finding not only shed light on the assembly process of the HIV-1 capsid but also highlighted its role in modulating post-assembly functions and interactions with antiviral agents like Lenacapavir. Similarly, Stacey et al. [76] further dissected the structural complexities of the HIV-1 capsid. By using SPA, they revealed two structural switches in the CA that regulate capsid curvature and host factor binding, providing insights into the evolutionary advantage of the capsid conical morphology and its implications for cell entry mechanisms. The potential of SPA in viral structural analysis extends beyond HIV-1. Sugita et al.’s study on the Ebola virus nucleocapsid [77] offers a prime example. By achieving a near-atomic resolution structure of the recombinant Ebola virus nucleoprotein–RNA complex, their research provided a detailed view of the virus’s genome encapsulation and the dynamic transitions between RNA-bound and RNA-free states. This study elucidated the nucleocapsid assembly mechanism and highlighted key structural features for potential antiviral drug development.

Moreover, Lazic et al.’s exploration of integrated differential phase contrast scanning transmission electron microscopy (iDPC-STEM) marked a technological advancement in SPA [78]. Other approaches, such as cryo-electron ptychography, also explored the possibility of the STEM being used in the field of SPA [79]. Demonstrating the efficacy of iDPC-STEM, they successfully obtained high-resolution structures of tobacco mosaic virus (TMV) at 3.5 Å. Despite not exceeding the resolution of that of commonly used CTEM, iDPC-STEM holds the potential to significantly reduce radiation damage during data collection. The scanning mode in STEM deposits electrons sequentially one pixel at a time, which allows energy to spread and dissipate towards nonilluminated areas, potentially weakening the impact and damage at exposed spots. It has been shown that spot scanning of 100 nm beams in CTEM mitigates beam-induced motion and improves contrast. In addition, alternative scan patterns can further reduce beam damage. This advancement opens new possibilities for achieving even higher resolution structures with cryo-EM, further pushing the boundaries of our understanding of viral architectures.

## 6. Conclusions and Outlook

Advancements in multiscale imaging techniques have significantly enhanced our understanding of RNA virus biology, shedding light on the highly dynamic interactions between viruses and their hosts, as well as the complex cellular responses. Whole-cell imaging techniques offer valuable insights into the ultrastructural alterations induced by viral invasion, while high-resolution imaging techniques provide detailed delineation of molecular mechanisms. On the other hand, RNA virus research and the development of multi-modal imaging methods are highly intertwined, as virus research has also contributed to the advancement of imaging techniques. For instance, the study of HIV-1 capsids has led to the development of a set of tools with extensive implications for in situ structural biology by bringing cryo-ET to subnanometer and near-atomic resolution [59,80,81,82].

Looking ahead, continued advancements and emerging trends hold great promise for characterizing viral infections from a holistic perspective. Ongoing and future efforts are essential to bridge the gap between “in vitro” and “in cellular” RNA virus research. This includes integrating various imaging modalities, incorporating fluorescent microscopy methods for correlative imaging approaches to enhance precision in targeted imaging, and refining reconstruction algorithms by leveraging artificial intelligence (AI) and machine learning. Enhancing accessibility, automation, and a diverse array of approaches adaptable to various scales will offer crucial tools for developing novel antiviral strategies and mounting rapid responses to emerging viral threats.

## Figures and Tables

**Figure 1 viruses-16-01504-f001:**
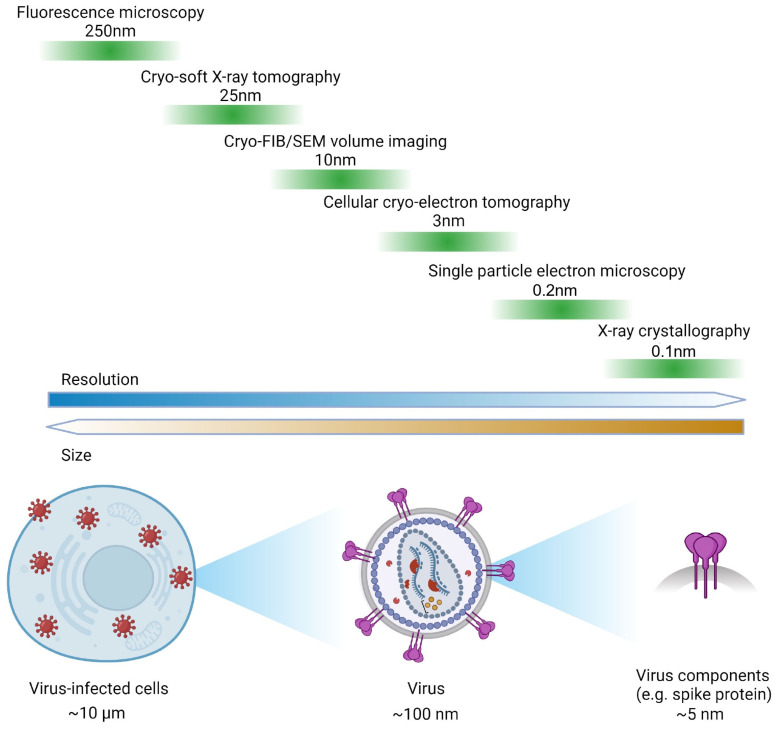
Schematic diagram showing the resolution capabilities of various imaging techniques in the research of RNA virus. Upper: resolution ranges of various imaging and structural methods. Lower: Schematic representations of the size scales of virus-infected cells, virus particles, and virus components (e.g., spike proteins). Different approaches can be utilized depending on the scale of the research object.

**Figure 2 viruses-16-01504-f002:**
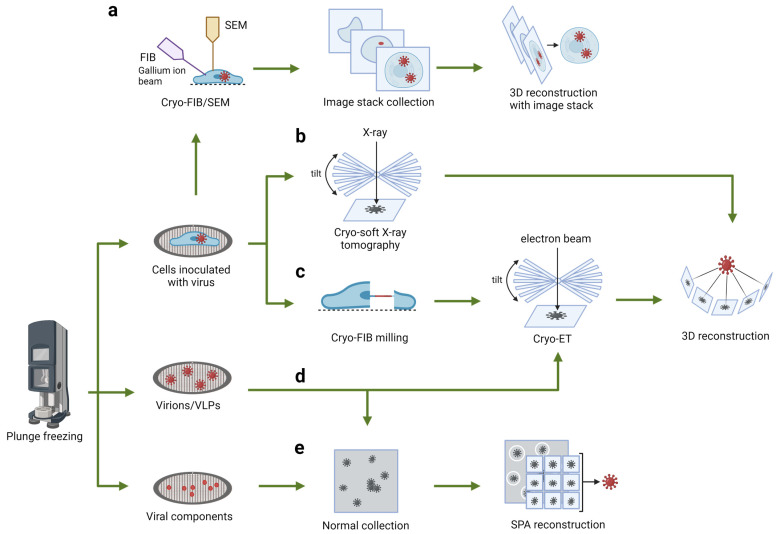
General workflow for microscopical research on RNA viruses. Samples are applied to grids and undergo a vitrification process (the most well-known procedure being plunge freezing), with subsequent imaging procedures depending on the sample. (**a**,**b**) For samples containing whole mammalian cells, serial cryo-focused ion beam/scanning electron microscopy (cryo-FIB/SEM) (**a**) and cryo-soft X-ray tomography (cryo-SXT) (**b**) can be used for volume imaging. (**c**) To achieve high-resolution imaging of regions of interest (ROI) within the cell (e.g., virus particles), a thin lamella of 80–250 nm can be prepared using FIB milling and imaged using cryo-ET. (**d**) Virions and VLPs can be imaged and structurally characterized using both cryo-ET or SPA. (**e**) For purified viral components (e.g., spikes), SPA is often the preferred method for structural analysis.

**Figure 3 viruses-16-01504-f003:**
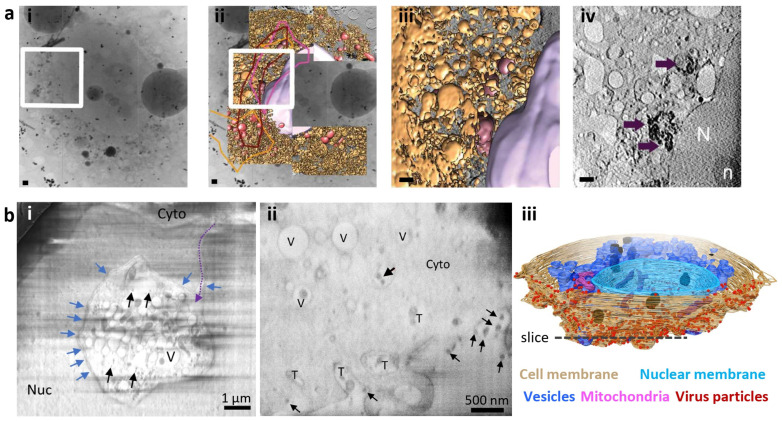
Cryo-SXT and cryo-FIB/SEM techniques are suitable for imaging and reconstructing intact virus-infected cells. (**a**) Three-dimensional reconstruction of whole-cell volumes of HCV replicon-bearing cells by cryo-SXT (scale bars, 1 µm) [9]. (**i**,**ii**) Tile-scanned projection (**i**) and overlaid with segmentation (**ii**) showing the area selected for SXT (white square); The enlarged view of segmented volume (**iii**) and tomographic slice (**iv**) from the boxed areas are also shown. Threshold-based isosurface segmentation of the surface boundaries identifies the different organelles present in the cells: abnormal mitochondria in red and purple, ER in yellow, modified ER in brown and nuclear envelope in pink. Purple arrows point to the abnormal mitochondria. (**b**) Characterization of SARS-CoV-2-infected Vero cells using cryo-FIB/SEM volume imaging [10]. Black arrows, SARS-CoV-2 virus particles; blue arrows, nuclear pores; dashed purple arrow, invagination path; Nuc nucleus; Cyto cytoplasm; V vesicles; T tunnels. (**i**) A representative cryo-FIB/SEM slice showing the invagination of cytoplasm into the nuclear space (scale bar, 1 µm). (**ii**) A representative cryo-FIB/SEM slice showing the distribution of SARS-CoV-2 particles (scale bar, 500 nm). (**iii**) Surface rendering of the segmented volume of a SARS-CoV-2-infected cell as shown in (**ii**). Segmented organelles and virus particles are labeled with indicated colors. The dashed lines indicate the position of the slice shown in (**ii**).

**Figure 4 viruses-16-01504-f004:**
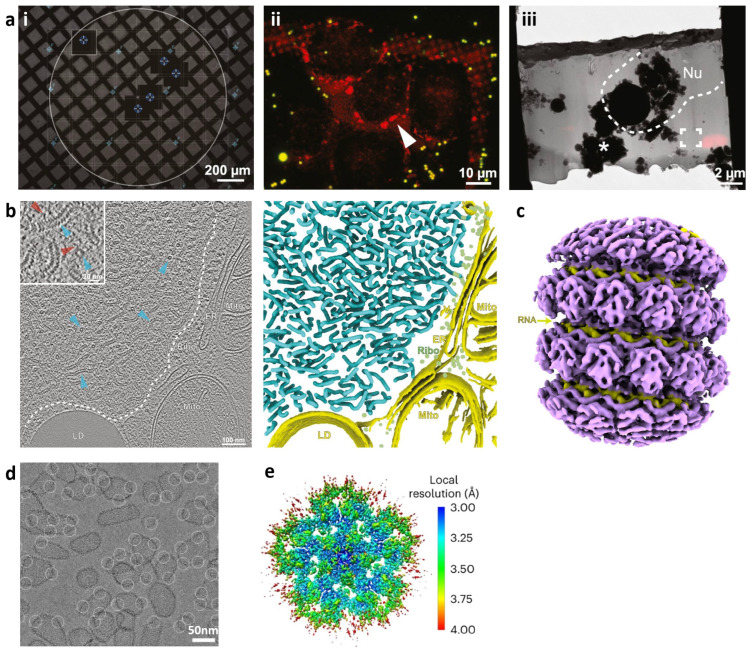
Cryo-ET and cryo-EM for high-resolution characterization of viral structure in situ. (**a**) A cryo-correlative light and electron microscopy (CLEM) workflow targeting unlabelled MuV near fluorescently labelled mCherry-G3BP1 stress granules (SGs) [11]. (**i**) MuV-infected HeLa cells were seeded on an EM grid and imaged by phase contrast light microscope (scale bar, 200 µm). (**ii**) Infected cells with red fluorescent SGs (white arrowhead) are targeted for FIB-milling (scale bar, 10 µm). (**iii**) Lamella TEM map overlaid with cryo-confocal images confirming the presence of SGs post-milling and guiding cryo-ET data collection (scale bar, 2 µm) (Nu: nucleus). (**b**) Representative tomographic slice (left) and corresponding 3D segmentation (right) for in-cell MuV characterization (scale bar, 100 nm). Side and top views of MuV nucleocapsids are marked with cyan arrowheads and colored accordingly. The inset shows nucleocapsids have extended flexible densities (red arrowheads). Both the tomographic slice and segmentation are labelled with lipid droplets (LD), mitochondria (Mito), endoplasmic reticulum (ER), ribosomes (Ribo), and vesicles (V) [11]. (**c**) Subtomogram average of in-cell MuV nucleocapsids resolved to 6.5 Å (EMD-13137), with RNA densities in yellow. (**d**) Representative micrograph of in vitro assembled HIV-1 capsids with picked pentamer particles (white circles) (scale bar, 50 nm) [12]. (**e**) High-resolution reconstruction of HIV-1 capsid pentamer structure colored according to local resolution (EMD-28186) [12].

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
