# Peer review of "Zooming in and out: Exploring RNA Viral Infections with Multiscale Microscopic Methods"

_viruses, 2024, doi:10.3390/v16091504_

Round 1

Reviewer 1 Report

Comments and Suggestions for Authors

This is an interesting and comprehensive review describing various advanced microscopic approaches that help to investigate the interaction of the RNA viruses with the host cell and reveal the fine details of the structure of viral proteins and the molecular mechanisms of virus assembly. It provides enough excellent examples to illustrate every technique described and showing advantages of using sample freezing (several variants of cryoelectron microscopy/tomography, sometimes combined with light/ fluorescent microscopy, including focused ion beam scanning microscopy).

My criticism mainly concerns the design of the manuscript:

(1) The abbreviations of the techniques described are not always provided at their first use. So, only after a long search did I find in line 132 the decoding of the abbreviation cryo-FIB/SEM (cryo-focused ion beam/scanning electron microscopy), which had been used many times before, including the Abstract and Keywords. One abbreviation - GIS - depicted in Figure 1a was not decoded at all.

(2) Mentioning the Figures in the text is often inconsistent, and it's a little annoying. Thus, in Section 2 the authors start with Figure 2b (line 84), and continue with Figure 3a in the next paragraph (line 96) while Figure 2a is first discussed much later in Section 3 (Line 137), and Figures 2c and 2d appeared first in Section 4 (lines 189 and 194, respectively), and Figure 2e appeared in line 261. The same discrepancy is observed when some parts of Figure 3 are mentioned in the text. It would be much easier to follow the authors' thought if they refer to parts of the Figures in a more orderly way.

(3) Finally, I did not find any mention in the Figure legends about whether those images were published earlier (and where). I think it is necessary to provide some references to the original papers.

Author Response

This is an interesting and comprehensive review describing various advanced microscopic approaches that help to investigate the interaction of the RNA viruses with the host cell and reveal the fine details of the structure of viral proteins and the molecular mechanisms of virus assembly. It provides enough excellent examples to illustrate every technique described and showing advantages of using sample freezing (several variants of cryoelectron microscopy/tomography, sometimes combined with light/ fluorescent microscopy, including focused ion beam scanning microscopy).

My criticism mainly concerns the design of the manuscript:

(1) The abbreviations of the techniques described are not always provided at their first use. So, only after a long search did I find in line 132 the decoding of the abbreviation cryo-FIB/SEM (cryo-focused ion beam/scanning electron microscopy), which had been used many times before, including the Abstract and Keywords. One abbreviation - GIS - depicted in Figure 1a was not decoded at all.

We appreciate the reviewer’s comments. We have ensured that all abbreviations are presented in their entirety upon first mention throughout the abstract, keywords, figures, and main text. These have been highlighted for clarity. Additionally, we removed GIS in Figure 2a to prevent any potential confusion.

(2) Mentioning the Figures in the text is often inconsistent, and it's a little annoying. Thus, in Section 2 the authors start with Figure 2b (line 84), and continue with Figure 3a in the next paragraph (line 96) while Figure 2a is first discussed much later in Section 3 (Line 137), and Figures 2c and 2d appeared first in Section 4 (lines 189 and 194, respectively), and Figure 2e appeared in line 261. The same discrepancy is observed when some parts of Figure 3 are mentioned in the text. It would be much easier to follow the authors' thought if they refer to parts of the Figures in a more orderly way.

Thank you for bringing this to our attention. To improve the manuscript's flow, we have reordered the figure citation appearance to make it consistent with the text. We also added a clarifying sentence in the introduction to explain the logic behind the figure references. Specifically, we organized the figures to reflect the progressive nature of the techniques discussed. Figure 1 summarizes the achievable resolutions of each technique, Figure 2 illustrates their respective workflows, and Figures 3 and 4 cover applications.

(3) Finally, I did not find any mention in the Figure legends about whether those images were published earlier (and where). I think it is necessary to provide some references to the original papers.

We have now added references to the figure legends where appropriate, citing the original papers for any previously published images.

Reviewer 2 Report

Comments and Suggestions for Authors

RNA viruses, consisting of single-stranded (ssRNA), double-stranded (dsRNA) and retroviruses, include a number of important human pathogens such as influenza, SARS-CoV-2, Ebola and human immunodeficiency virus to name just a few.  All are characterized by particularly high mutation rates, which enable them to relatively easily escape immune defense mechanisms.  This, in turn, renders the development of effective vaccine and therapeutic strategies extremely difficult.  For these reasons, a detailed understanding of all aspects of virus structure, the virus life cycle and virus interactions with the host cell is vital to enable the development of efficacious countermeasures for these viruses.  While biochemical and genetic approaches have, and will continue to be, informative, cryo-imaging techniques are poised to significantly enhance our understanding of each of these aspects of the virus and its interaction with cellular components. 

The authors do an outstanding job of differentiating between the various cryo-techniques organized according to observation scale from large to small, including cryo-soft X-ray tomography, serial cryo-FIB/SEM volume imaging, cellular cryo-tomography and single particle cryo-electron microscopy.  The comparative resolution capabilities of these technologies are perfectly depicted in Fig. 1.  The review then goes on to discuss in exquisite detail some of the experimental aspects, capabilities and published findings for each technique.  Convincing images are then included to illustrate the power of each technology.

There are no discernible weaknesses or omissions in the manuscript.  It is considered a timely, accurate and highly detailed treatise on the power and scope of cryo-imaging technologies and, thus, constitutes a major contribution to the field. 

Author Response

Thanks very much for the positive comments from the reviewer.

Reviewer 3 Report

Comments and Suggestions for Authors

The authors reviewed the innovative multiscale microscopic methods used to study RNA viruses, ranging from fluorescence microscopy to advanced techniques like cryo-electron tomography. By combining these methods, researchers can explore the intricate structures and dynamics of RNA viruses, enhancing our understanding of their interactions with host cells. Generally, it is a good review and focused on the frontier of the field. It is a well written review and I think it could be published in this form.

Author Response

We thank the reviewer for the positive comments.